# Arming Filamentous Bacteriophage, a Nature-Made Nanoparticle, for New Vaccine and Immunotherapeutic Strategies

**DOI:** 10.3390/pharmaceutics11090437

**Published:** 2019-09-01

**Authors:** Rossella Sartorius, Luciana D’Apice, Antonella Prisco, Piergiuseppe De Berardinis

**Affiliations:** 1Institute of Biochemistry and Cell Biology (IBBC), 80131 CNR Naples, Italy; 2Institute of Genetics and Biophysics “A. Buzzati-Traverso” (IGB), 80131 CNR Naples, Italy

**Keywords:** filamentous bacteriophage, vaccine, nanoparticle, targeting, phage display, antigen delivery

## Abstract

The pharmaceutical use of bacteriophages as safe and inexpensive therapeutic tools is collecting renewed interest. The use of lytic phages to fight antibiotic-resistant bacterial strains is pursued in academic and industrial projects and is the object of several clinical trials. On the other hand, filamentous bacteriophages used for the phage display technology can also have diagnostic and therapeutic applications. Filamentous bacteriophages are nature-made nanoparticles useful for their size, the capability to enter blood vessels, and the capacity of high-density antigen expression. In the last decades, our laboratory focused its efforts in the study of antigen delivery strategies based on the filamentous bacteriophage ‘fd’, able to trigger all arms of the immune response, with particular emphasis on the ability of the MHC class I restricted antigenic determinants displayed on phages to induce strong and protective cytotoxic responses. We showed that fd bacteriophages, engineered to target mouse dendritic cells (DCs), activate innate and adaptive responses without the need of exogenous adjuvants, and more recently, we described the display of immunologically active lipids. In this review, we will provide an overview of the reported applications of the bacteriophage carriers and describe the advantages of exploiting this technology for delivery strategies.

## 1. Introduction

Bacteriophages only infect and multiply with their specific host and currently, as bacterial resistance to antibiotics becomes widespread, the therapeutic use of bacteriophages is back on the agenda [1]. Recently, on the basis of successful treatments in five people with phage cocktails under a U.S. food and drug administration (FDA) process designed for emergencies, the University of California, San Diego (UCSD) is launching the Center for Innovative Phage Applications and Therapeutics (IPATH) to refine phage treatment and perform phage trials focused on patients with a single, known bacterial infection and without withholding other potentially beneficial treatments, including antibiotics [2]. In this context, an efficacious clinical treatment of *Mycobacterium abscessus* infection in a 15 years-old cystic fibrosis patient using a cocktail of three genetically engineered phages was recently reported [3], re-launching the use of bacteriophages as nanopharmaceuticals against antibiotic-resistant bacteria.

Besides the use of lytic phages as antibacterials, which has been extensively reviewed elsewhere [2], a collection of evidence was obtained in recent years on the potential translational usage of filamentous bacteriophages.

Filamentous phages are single-strand DNA virions belonging to the Inoviridae family; phages f1, fd, and M13 are a sub-group of rod-like shaped Inoviruses with a repeated and ordered capsid structure. They are closely related species, sharing almost the same genome (with only about 1–2% of difference) that infect and replicate in *Escherichia coli* bacterial cells and which are often collectively referred to as Ff phages.

Their peculiar proteic structure, together with the flexibility of the DNA genome and the easiness of purification has fostered their application in the phage display technology, with particular attention to the production of therapeutic antibodies first, and then as antigen delivery system for the development of new vaccine formulations [4,5].

Although practically every filamentous bacteriophage coat protein can be used to display foreign amino acidic sequences, the pVIII protein is the most used for the expression of exogenous peptides in high copy number. The pIII protein, instead, has been successfully used for the expression of up to five copies per virion of the receptor-ligand and single-chain antibody fragment (scFv) [6] (Figure 1). 

The foreign sequence expression is highly stable since the (poly)peptides of interest are genetically fused to phage proteins and not linked by chemical bonds. Small peptides can be easily expressed on all copies of the pVIII major coat protein, whereas for the expression of longer sequences (14–20 amino acids), such as immunologically relevant peptides, it is often necessary to construct hybrid phages, which express recombinant copies of the pVIII interspersed with wild-type copies, in order to guarantee a high expression of the exogenous sequence without affecting the stability of the phage lattice. Due to the highly-symmetrical and repeated structure, Ff phages are suitable for the high-density exposure of one or more epitopes on the coat surface. Overall phage nanoparticles are made of highly organized monomers represented mainly by the pVIII protein (5.5 kDa), an alpha-helix closely packed to compose a right-handed helical latex with a rod-like structure.

Combining a relatively simple surface structure with a particulate nature and with adjuvant properties, filamentous phages represent ideal nanoparticles compared to other carriers. Drugs can be easily conjugated to the phage surface by chemical modification of amino and carboxyl groups exposed in the amino terminus of the pVIII protein. Furthermore, the filamentous phages are extremely tolerant to variations in the size of their genome, which makes them versatile for engineering through the phage display technique, allowing the successful expression of B and T (CD4+ and CD8+) epitopes on the phage nanoparticles, and between the end of the 1990s and the first decade of 2000, filamentous phages were widely tested as immunogenic nanocarriers both in vitro and in vivo [6,7,8].

A number of scientific reports discuss the use of filamentous bacteriophage nanoparticles as a delivery system for drugs, as reported elsewhere [9]. The aim of this review is to give an overview of the recent literature on the use of filamentous bacteriophage nanoparticles as a delivery system for vaccine and as immune response inducer (As literature search parameters, we at first performed a search to identify articles with keywords *filamentous bacteriophage and phage-display and vaccines, filamentous bacteriophage and cytotoxic response, filamentous bacteriophage and cellular response, filamentous bacteriophage and antibodies,* in the title, abstract, keywords, topic. Representative publications regarding phage-based vaccines and immunotherapeutic strategies were selected and then citing publications were searched, to identify recent developments. Quoting of reviews was limited and we left out articles on the same topic from the same author). In addition, and more thoroughly, we report work performed by us in the last decades using filamentous bacteriophage fd, illustrating the efficacy of this versatile tool and showing our recent findings and upgrading of the bacteriophage fd system for translational immunotherapeutic applications.

## 2. The Use of Filamentous Bacteriophages for the Induction and Analysis of Antibody Response

Filamentous bacteriophages can be utilized both to analyze the specificity of antibody responses and to induce a humoral immune reaction. In some instances, filamentous phages have been utilized to integrate epitope discovery and immunization functions into a single platform.

Precise determination of conformational epitopes of neutralizing antibodies represents a crucial step in the rational design of novel vaccines; the screening of random peptide libraries displayed on phages, and of gene or genome fragment phage libraries provides a powerful, cheap and quick technique for epitope mapping [10]. Phages bearing peptides that reproduce an epitope of interest can be selected from random peptide phage-display libraries, displaying more than 10^9^ different peptides, utilizing monoclonal or polyclonal antibodies; in some cases, the selected peptide-displaying phage can then be used as an immunogen, to induce an antibody response [11,12]. Filamentous bacteriophage M13 is most commonly used to generate combinatorial libraries. Phages selected by phage library biopanning rarely display peptides that are identical in their primary sequence to a peptide of the protein antigen of interest; the method more commonly leads to the identification of mimotope peptides. A mimotope peptide can mimic both a linear epitope and a discontinuous conformational epitope [13]; it can also mimic a non-peptidic epitope, such as a lipopolysaccharide epitope or a glycan [14,15]. Genome-fragment phage display libraries are collections of phages engineered to express short polypeptides of a pathogen’s proteome. Phage libraries that contained fragments of the genome of the H5N1 strain responsible for an outbreak of human influenza in Vietnam in 2004–2005 were used to analyze the antibodies that made people recover from the infection [16]. The insert sizes, in the pIII gene, ranged between 50–200 base pairs (bp) and 200–1000 bp. This strategy allowed the identification of noncontinuous conformation-dependent epitopes—protein sequences that are not adjacent to one another in the polypeptide sequence of the protein, but that lie close together in space in the folded protein [16]. Moreover, whole-genome-fragment phage display libraries followed by surface plasmon resonance technologies were used to elucidate the effect of different adjuvants on the antibody repertoire against an H5N1 vaccine in humans [17]. Whole-genome phage display library spanning the entire Zika virus genome has been used for in-depth immune profiling of IgG and IgM antibody repertoires in longitudinal serum and urine samples from individuals acutely infected with Zika virus [18]. Furthermore, phage-displayed peptide libraries have been utilized to identify B epitopes on allergens, pathogens, and human proteins [19,20,21,22,23]. Thus, phage libraries are invaluable tools for the epitope specificity of antibody response analysis. In some cases, the phages selected from these libraries have been tested as vaccines. 

The use of filamentous phages for the induction of antibody responses has been tested by researchers involved in the generation of vaccines against fungal infections. There is still no approved antifungal vaccine or antibody for use in humans; it is clear that humoral and cellular immunities are the most important host defense mechanisms against fungal infections; severe infections mainly occur in immunocompromised patients. In mice, vaccination with filamentous bacteriophages displaying a peptide from Fructose-bisphosphate aldolase1 (Fba1 epitope YGKDVKDLFDYAQE) of *Candida albicans* induced humoral and cellular immune responses, reduced fungal burden, and relieved kidney damage in infected mice and significantly improved their survival rates. However, the protective efficacy of the phage vaccines was lower than the efficacy of vaccination with the whole Fba1 protein [24].

Another phage-based vaccination strategy against *Candida albicans* focused on heat shock protein (HSP) 90, a protein that plays a key pathogenic role in systemic infection. Hybrid-phage particles expressing the HSP90 DEPAGE epitope induced the specific antibody against HSP90, enhanced the cellular immune responses in mice, and afforded some protection from systemic candidiasis; hybrid-phage-immunized mice had fewer Colony-Forming Units in the kidneys compared with wild-type-immunized mice and vehicle-injected mice and had a statistically significant survival advantage over vehicle-injected group [25]. Furthermore, an fd bacteriophage functionalized with peptides from secreted aspartyl proteinase (Sap) 2 and Hsp90 was designed to capture or induce anti-Sap2 IgG and anti- Hsp90 IgG simultaneously since anti-Sap2 antibodies were found to be protective in people with systemic candidiasis [26].

Vaccination with filamentous bacteriophages has also been tested against another fungal pathogen, *Sporothrix globosa*. In this case, an epitope peptide (sequence KPVGHALLTPLGLDR) derived from a 70 KDa glycoprotein of *S. globosa,* that can induce a protective response, was displayed on the major coat protein pIII. Immunization with recombinant phage increased the survival rate of mice following *S. globosa* infection. The nature of the mechanisms underlying the resolution of infection in mice treated with recombinant phage is unresolved; it can be hypothesized that protective antibodies are elicited, which also increases the cell-mediated immune response [27].

Filamentous phages fd as nanocarriers of B cell epitopes proved useful for the induction of antibody responses to β-amyloid peptide, which is the main component of the plaques present in the brain of Alzheimer’s Disease (AD) patients. Immunization studies performed in transgenic mouse models of β-amyloid deposition have demonstrated that antibodies against β-amyloid can reduce amyloid load and improve cognition. The N-terminus of the beta-amyloid peptide is considered the most promising antibody target for inclusion in recombinant vaccines [28,29]. Antibody 6C6, a mouse monoclonal that disaggregates β-amyloid fibrils in vitro, was used to screen a random 15-mer peptide phage library, leading to the selection of several phages displaying the EFRH sequence, a B cell epitope corresponding to amino acids 3–6 within the human β-amyloid peptide [30]. Upon immunization, the phages displaying sequence EFRH induced antibodies with the same disaggregating properties as 6C6 [30]. Subsequent studies investigated the immunogenicity of phages that displayed varying numbers of copies of the EFRH epitope on the phage nanoparticles.

In particular, Solomon and collaborators generated phages containing around 150 recombinant copies of pVIII protein out of the 2700 copies of pVIII. The recombinant pVIII was modified to display either the EFRH sequence or the tandem repeat EFRHEFRH. The phages bearing the tandem repeat, and thus around 300 copies of epitope EFRH per nanoparticle, induced higher antibody titers than phages expressing 150 copies of the epitope, an observation that highlights the relevance of epitope density in phage immunizations [31]. Importantly, AD model mice treated with phages that express the EFRH epitope show a considerable improvement in their amyloid load and cognitive behavior [31,32].

We compared the immunogenicity of four different portions of the beta-amyloid peptide, displayed on filamentous phages fd, and observed that sequence AEFRH (corresponding to amino acids 2–6 within the human β-amyloid peptide) was the most immunogenic. As the first two amino acids of the processed N-terminus of the pVIII protein are an alanine (A) and a glutamic acid (E), we inserted in the pVIII only three extra amino acids, namely sequence FRH, to obtain sequence AEFRH. This strategy afforded the display of epitope 2–6 (AEFRH), at a density of 810 copies per phage particle [4,33]. AD model mice immunized monthly with AEFRH phages from age two months displayed a significant reduction in the number of β-amyloid plaques in the hippocampus and cortex at eight months. We observed however that the dosing protocol strongly affected the efficacy of vaccination with AEFRH phages; therefore, we hypothesize that the titer and affinity of the induced antibodies are affected by the dosing protocol and are crucial for efficacy [33,34,35,36]. Overall, the analysis of different epitopes of beta-amyloid suggests that the intensity of antibody response depends on characteristics of the displayed peptide, the method of its display and the dosing protocol. An epitope displayed in a high copy number on the surface of a phage is more effective in eliciting high antibody titers than the same epitope displayed in low copy number. A potential problem in single epitope anti-beta-amyloid vaccines is inter-individual variability in anti-beta-amyloid antibody titer and in the development of immunological memory to the B epitope of interest [37,38]. In this context, we are currently investigating the effect of the dosing protocol on these immunologic outcomes. We hypothesize that vaccines containing several epitopes may mitigate the problem of inter-individual variability in the immune response.

Overall, filamentous phages proved very useful for the analysis of antibody specificity. As regards their use as immunogens, the data in the literature show that phage-displaying B cell epitopes in high copy number on the pVIII coat protein are often able to induce an antibody response against the displayed antigenic determinant. One problem, however, is that the induction of an antibody response against a single peptide does not afford protection. Moreover, inter-individual variability in titer and memory may prove a general roadblock in the development of vaccines that focus the antibody response against a single B cell epitope; further research is required to address this problem, and the use of mixtures of phages displaying different epitopes being one of the ways to overcome it.

## 3. Filamentous Bacteriophage Nanocarriers for the Induction of Cellular Immune Responses

Hybrid filamentous virions expressing T helper (Th) peptides derived from HIV-1 (epitope Pep23, corresponding to residues 249–263 of the RTase) [39,40] or Human Cytomegalovirus (HCMV epitopes p128 and p30 of Pp65 protein) [41] as a fusion with the major coat protein pVIII were recognized by specific CD4+-restricted human T-cell lines and T cell clones and enhanced peptide antigenicity. Moreover, as mentioned in the previous paragraph, B and T cell epitopes from *C. albicans* HSP90 protein exposed on the pIII protein of recombinant bacteriophages and administered to mice, induced antibodies as well as cell-mediated immune responses and prolonged mice survival after in vivo systemic candidiasis challenge [25,42]. The protective response induced by the *Candida* antigen exposed on the phage surface was manly Th1 and Th17 oriented, with the production of cytokines like IL-2, IL-12, IFN-γ, and IL-17, conferring resistance to most fungal pathogens [24,43].

Indeed, the use of bacteriophages as nanocarriers of antigenic peptides demonstrated effective mainly to the induction of specific cytotoxic T cell responses.

Several cytotoxic T lymphocytes (CTL) epitopes were displayed on the filamentous phage lattice. Bacteriophages were shown to be effective in triggering a cytotoxic response to HLA-A2 restricted epitope ILKEPVHGV from HIV-1 reverse transcriptase (residues 309–317) [44,45] and Hepatitis B virus epitope S_28–39_ [46]. Mascolo et al. reported that the CTL response evoked in mice by administration of recombinant bacteriophage appeared to be mediated by the induction of T CD4+ lymphocytes stimulated by Th epitopes contained in the proteins of the phage capsid itself [47]. However, even if the co-display of an engineered helper peptide proved dispensable for the primary short-lived cytotoxic response, it was required for the induction of long-term memory CTLs [48].

It can be hypothesized that the success of filamentous bacteriophages as a nanodelivery system to induce cell-mediated responses against high-density expressed antigens is mainly due to the ability of the filamentous rod to be internalized and processed by antigen-presenting cells (APCs), and to the presentation of the displayed peptides in association with both MHC class II and MHC class I molecules [41,49]. Using macrophages to study recombinant phage particle intracellular fate, Wan et al. [50] demonstrated by confocal microscopy that the MHC class I-peptide exposed on bacteriophages are translocated from endosomes to ER during phagocytosis, suggesting that the mechanism of cross-presentation of phage particles occurs within the endocytic pathway: Endocytosed phage particles are transferred to the proteasome-mediated degradation pathway and peptides are then re-imported into phagosomes via the TAP complex for loading onto MHC class I molecules.

Phage-based nanovaccines have also been formulated for cancer treatment and prevention. The tumor-specific epitope 161–169 (EADPTGHSY) from melanoma-associated antigen (MAGE) A1 was successfully expressed on the surface of bacteriophage fd and used in vivo in a mouse model. MAGE-A1 bacteriophage injection elicited EADPTGHSY-specific cytotoxic T lymphocytes (CTL) responses and NK activity, protecting mice from tumor progression both in therapeutic and prophylactic schedule [51]. Similarly, phage expressing tumor antigen P1A_35–43_ prevented or suppressed tumor growth in a P815 murine mastocytoma model promoting CTL and Th1-oriented cellular responses [52]. Immunization with double hybrid filamentous bacteriophages co-expressing a tumor peptide from MAGE-A10 (amino acids 254–269) or MAGE-A3 (amino acids 271–279) together with a Th peptide of viral origin (Pep23) protected humanized HHD (HLA-A2.1/H2-Db) transgenic mice from tumor growth. In addition, human anti-MAGE-A3_271–279_-specific T cell clones isolated from bacteriophage-stimulated T cell lines showed high avidity for the MAGE-A3 epitope and were able to kill human MAGE-A3-tumor cell lines expressing a low amount of MAGE-A3_271–279_ peptide-HLA complex on the surface [53]. Immunization with a combination of wild-type bacteriophage fd in the presence of the Th and MAGE-A3_271–279_ CTL synthetic peptides did not induce CTL responses, demonstrating the ability of this nanocarrier to enhance the immunogenicity of the displayed peptides.

More recently, Roehnisch et al. developed an M13 bacteriophage nanovaccine against multiple myeloma (MM) and tested it in vivo in a phase I/II trial on human patients in the terminal stage of MM [54]. Bacteriophages chemically linked to idiotype (Id) proteins isolated from patients were generated and used to induce tumor-specific immune responses against MM cells. The results showed that intradermal immunization with Id-bacteriophages induced reduction of paraprotein levels in the blood, a symptom of responsiveness to the phage vaccination. Besides, several patients developed not only anti-Id antibodies but also a cellular response, as demonstrated by the positive delayed-type hypersensitivity (DTH) reaction induced in anti-Id positive patients. The DTH lesions revealed that phage administration in humans induces infiltration of neutrophils, macrophages, and CD8+ T cells. Moreover, peripheral blood mononuclear cells (PBMCs) from vaccinated patients were able to kill specifically myeloma target cells isolated from the bone marrow of the same patients. The phage vaccination was well tolerated by all the trial participants, demonstrating the safety and the efficacy of bacteriophage nanovaccines.

The filamentous bacteriophage delivery system can overcome the induction of immune tolerance due to self-antigens in cancer. Anti-HER2 phage-based vaccination breaks self-tolerance and can delay tumor progression in HER2 transgenic mice, inducing not only a sustained humoral response but also a massive infiltration of CD3+ cells in the tumors [55]. This phenomenon could be due to the high self-immunogenicity of the carrier. The delivery of bacteriophages into the tumor bed using the display of tumor ligands on the phage scaffold strongly induces macrophages and neutrophils recruitment and the release of Th1 cytokines that promotes the proinflammatory environment and activates immune responses toward cancer cells [56,57]. Carcinoembryonic antigen [CEA]-targeted M13 bacteriophage in vivo injected in mice implanted with colorectal tumors induced tumor infiltration of both neutrophils and tumor-associated macrophages (TAM), maturation of dendritic cells (DCs), and strong CD8+ T cell-mediated antitumor responses [57].

These mechanisms were demonstrated to be MyD88-dependent [58] and regulated by Toll-like receptor 9 (TLR9) [59] via unmethylated CpG motif contained into the single-stranded DNA genome of the virion [60]. Recently, Gomes-Neto et al. have shown that the in vivo vaccination with filamentous bacteriophages fd expressing PA8 and TSKB20 CD8+ epitopes from human protozoan *Trypanosoma cruzi* induced both humoral and cytotoxic responses and protected mice against parasitemia and parasite burden induced by a high-dose inoculum of *T. cruzi* parasite [61]. This effect was completely abolished in TLR9^-/-^ mice, reinforcing the idea that bacteriophages induce cellular and humoral responses through a TLR9-dependent mechanism.

### 3.1. Use of Filamentous Bacteriophages for Cell Targeting

The most efficient way to mount a sustained immune response is to deliver the antigens specifically to APCs to trigger both innate and adaptive immunity. This mission can be accomplished by the co-administration of both antigen and adjuvant to activate defined subsets of professional APCs such as DCs. Simultaneous delivery of antigen and adjuvant to the same antigen-presenting cell allows only cells internalizing the antigen to receive the proper activation signal, avoiding the induction of T-cell anergy due to the antigen presentation in the absence of co-stimuli.

A much-wanted option is thus to produce new generation vaccines able to combine pattern recognition receptors (PRR) ligands with antigenic proteins and to deliver them via specific targeting to DCs. In this context, we recently showed that filamentous bacteriophages fd could be engineered to target mouse DCs. Phage scaffold proteins are very tolerant of genetic modification, and their flexibility allows the generation of multivalent bacteriophages displaying two or more different exogenous peptides/proteins with targeting functionality and/or immunogenic proprieties. Phage display of an scFv directed to the DC surface receptor DEC205 significantly enhanced the virions uptake by DCs and strengthened cytotoxic T cell response against the ovalbumin (OVA)_257–264_ epitope co-expressed on the phage coat [62]. DEC205 is an endocytic receptor that internalizes its ligands into clathrin-coated vesicles and drives them into the deep endosomes or lysosomes, enhancing the MHC-antigen presentation [63]. Targeting the fd virion to this receptor leads to an improved and faster internalization. In addition, mice vaccinated with the hybrid fd virions inhibited the growth of the OVA-expressing tumors. The active delivery of bacteriophages into DEC205+ DCs favors DC maturation, inducing up-regulation of costimulatory molecules, cytokines, and chemokines release and DC migration to peripheral lymph nodes where adaptive immune responses will be initiated (Figure 2).

More recently, we demonstrated that the increase of proinflammatory cytokines and type I interferon by DCs targeted via DEC205 bacteriophages was due to the receptor-mediated internalization of bacteriophages into LAMP-1-positive late endosomal and lysosomal compartments of the DCs, where the active form of TLR9 resides. Thus, the DEC205 targeting strategy, enhancing the virion uptake, optimizes the TLR9 activation by the unmethylated CpG motifs contained into the single-strand DNA phage genome, increasing the presentation efficacy of the DCs [64].

The active delivery of bacteriophages to a target tissue or cell population via phage display of ligands for specific receptors can enhance bacteriophage therapeutic efficacy; currently, multidisciplinary approaches are used to identify tumor-binding ligands or vessel-specific homing peptides, including in vivo panning in cancer patients for the discovery of tumor-binding antibodies and peptides.

While injected, untargeted bacteriophages are distributed throughout the body; phages expressing peptides or antibodies against a specific target organ accumulate at the binding site, allowing the recovery of phages with high binding capability. In vivo phage-display library panning in human patients with stage IV cancer, including breast, malignant melanoma, and pancreas tumors, led to the recovery of tumor binding phages from the surgically excised cancer tissue and to the identification of tumor-homing peptides and binding antibodies that are unique targets for an individual patient and useful in diagnostic procedures or for drug delivery in cancer treatment [65,66]. Intravenous infusions of bacteriophages do not cause allergy or other immediate or delayed severe adverse reactions, supporting the safety of filamentous bacteriophage applications in humans [66].

Proteomics-based strategies [67] and next-generation sequencing (NGS) for in-depth analysis of the phage libraries [68] have further enhanced the phage display technique for the identification of pharmacological and diagnostic tools. Combining the screening of synthetic scFv library expressing on phage with the use of an NGS platform permits to increase the numbers of identified scFv candidates, covering a wide range of epitopes on the target protein, to drastically reduce the numbers of rounds of selection and also to follow the in vitro evolution of virtually all variable CDR3 sequences during the panning process leading to identification of high affinity antibodies [68]. Table 1 shows the antigens discussed throughout the text and used to elicit immune responses.

### 3.2. Further Improvement by Coupling Immunologically Active Molecules to Filamentous Bacteriophages

Filamentous bacteriophages engineered for targeting of different tumors or tissue were used not only for the delivery of immunogenic epitopes as a fusion with the coat proteins but were also engineered for the targeted nanodelivery of therapeutic agents, drugs [69], fluorescent dyes for imaging and diagnostic [70], DNA cassettes [71], and siRNA [72].

Recently, the high-content of hydrophobic residues contained in the core domain of the pVIII protein [73,74] was exploited to promote the association of the surface major coat protein of the fd phage with the alpha-GalactosylCeramide (α-GalCer), a lipid stimulating invariant natural killer T (iNKT) cells [75]. It is likely that bacteriophage fd/α-GalCer conjugates are mainly internalized by DCs and the delivered α-GalCer is presented in association with the CD1d molecules expressed on the surface of DCs. α-GalCer-bacteriophages were demonstrated able to repeatedly stimulate iNKT to proliferate and to produce cytokines without the induction of iNKT anergy that is commonly found using the soluble lipid [76]. Moreover, therapeutic vaccination with recombinant bacteriophages functionalized with α-GalCer and decorated with a tumor-associated peptide increased the induction of antigen-specific CD8+ T cells and delayed tumor growth in mice, promoting the recruitment of tumor-specific CD8+ T cells inside the tumor bed [75].

## 4. Altered Gene Expression after Filamentous Bacteriophage Endocytosis

Due to their bacterial origin, bacteriophages can stimulate the innate immune response, inducing a continuous low level of immune reactions [77]. When used as antigen delivery vectors for vaccine formulation, filamentous bacteriophages are internalized by professional APCs and, once inside the cell, induce a substantial modification of the gene expression, as measured by transcriptome analysis [64,78]. Here we review recent findings obtained by RNA-Sequencing (RNAseq) which show unique adjuvant-like behavior of fd nanoparticles, revealed by significant perturbation of immune-related pathways.

Data obtained by RNAseq on DCs derived from the bone marrow of C57Bl/6 mice cultured with LPS-purified fd bacteriophages for 20 h showed significant gene deregulation: In a stringent analysis of differentially expressed genes, we reported that genes involved in the bacterial invasion pathway, TNF signaling pathway, and focal adhesion pathway were upregulated. At the very low level, the deregulation of genes involved in the Toll-like receptor pathway has also been observed [78]. Among the downregulated genes, we found the ones involved in the extracellular matrix (ECM) receptor interaction pathway and the protein digestion and absorption pathway. This is probably due to the internalization route followed by the bacteriophages: They enter mostly via pinocytosis and are directed into the cellular phagosomes and cytosol, where they alert intracellular danger signals starting gene expression modulation. Analysis of differentially expressed genes in bone marrow-derived DCs (BMDC) challenged with fd nanoparticles has been performed not only using the wild-type bacteriophage but also with a bacteriophage expressing a single-chain antibody fragment at the amino-terminus of the pIII protein, able to target the nanocarriers to the specific receptor DEC205. As previously reported [70], the effect of the receptor-targeted nanocarrier on gene expression modification is strongly increased compared to the wild type, and this is due to the localization of the former to an intracellular compartment embedded with TLR molecules able to trigger the innate immune response.

DEC205 mediated endocytosis indeed, drives the bacteriophages into the LAMP1+ compartments [79], where they meet the TLR9 molecule, and trigger the innate immune response. RNAseq of mouse bone marrow-derived DCs (BMDCs) pulsed with DEC205 targeted bacteriophage shows that the interaction of the virion with receptors of the danger response activates massive gene deregulation. Bacteriophages targeted DCs, compared with DCs treated with helminths or bacteria [80], show the same transcriptome reorganization, with massive upregulation of genes involved in the TLR pathway, the Cytokine–Cytokine receptor interaction, cytosolic DNA sensing, RIG-I like receptor sensing, and chemokine signaling. This demonstrates that the use of an antigen delivery system of prokaryotic origin can evoke an immune response able to induce DC maturation and start the downstream process of cytoskeleton reorganization, nuclear localization of transcription factors and cytokine production (Figure 3). Taken together, this altered gene expression is responsible for the innate immune response activation that is the prerequisite to induce a strong adaptive immune response and avoid tolerance.

Looking in detail at the genes involved in this impressive change in the transcription pattern, we found among the upregulated genes *MyD88* and *Unc93B1*, involved in the TLR downstream pathway and endosomal trafficking, respectively; *Psme1/2*, encoding the immune-proteasome associated complex PA28 subunits alpha and beta, where PA28 is a proteasome regulator and its expression is enhanced during the DC maturation process [81]. Two MHC-linked proteins, TAP1 and TAP2, required for the antigen-processing and presentation pathway of intracellular antigens to T cells show upregulated expression, as well, giving further cues that targeted bacteriophages drive dendritic cell maturation [82]. Of note, genes strongly transcribed (Table 2) are *Ifi203*, *Tmem173*, *Zbp1*, *Trex1,* and *Mb21d1* encoding the protein IFI203, STING, DAI, TREX1, and the cyclic GMP–AMP synthase cGAS, respectively. These proteins are responsible for cytosolic DNA sensing and degradation and they start the interferon-mediated response, with the downstream upregulation of several genes, culminating in dendritic cell maturation and cytokine and chemokine production. It is to be highlighted that more than 50% of upregulated genes are interferon regulated by IRF1, IRF2, IRF7, and interferon-stimulated responsive elements (ISRE) interspersed into their promoter. Notably, also the *Stat1, Stat2,* and *Irf9* genes are upregulated upon DEC205 targeted bacteriophage uptake. The STAT1/STAT2/IRF9 proteins form the heterotrimeric ISGF3 complex that binds the ISRE sequences and regulates the transcription of several genes [83]. Among these are the *Gbp2*, *Gbp3,* and *Gbp7*, all members of the same gene family, involved in the production of antimicrobial peptides and regulated by IFN, such as the member of *Ifit* family (inhibitors of viral replication) [84] and IFN-stimulated gene (ISG) family. Interferon activated genes are also the ones encoding for the OAS [85] and EIF2AK2 [86] proteins: Enzymes involved in blocking protein synthesis and DNA replication after viral infection, while the NRLP3 protein, one of the inflammasome complex, is involved in the production of active IL-1b and IL-18 cytokines after protease cleavage [87]. The downstream effects of the DC maturation after bacteriophage targeting are the expression of chemokine receptors and the chemokine secretion: CCR1, CCR2, CCR5, IFNAR1, IFNAR2 receptors, CXCL1, CXCL2, CXCL3, CXCL10 chemotactic chemokines, CXCL10, CCL7, CCL2, CCL3, CCL4, CCL5, CXCL12, IL-1A, IL-1B, IL-18, inflammatory molecules have been described as upregulated upon bacteriophage-mediated activation. The modification of the DC cytoskeleton, the overexpression of surface proteins, the secretion of chemoattractant and immune regulator molecules are the ultimate effects of these altered gene expressions.

Only a few genes are down-modulated and, among them, there are the ones involved in the oxidative phosphorylation (OXPHOS) pathway. These genes down-modulation is related to a metabolic switch first reported in tumoral cells and named, after the researcher who described this effect, as “Warburg”; more recently, it has also been described in innate immunity-related cells such as macrophages [88] and DC, where it has been related to the TLR mediated activation [89]. The Warburg effect is a modification of cell metabolism: In normal conditions, cells produce high amount of ATP molecules (36 ATP molecules from the complete oxidation of 1 glucose molecule) converting glucose in pyruvate and producing energy; the aerobic glycolysis or Warburg effect is the production of a high number of lactate molecules even in normal oxygen conditions. The lactate is the by-product of glucose metabolism, and it has been reported that this metabolic switch in aerobic conditions could be related to the need for an increase of fatty acid production for cell differentiation. One of the mediators of the Warburg effect is the transcription factor HIFalpha, encoded by the *Hif1A* gene, and also this gene is upregulated after bacteriophage uptake.

In conclusion, according to the data obtained by RNAseq, we propose the hypothesis that the DC-targeted fd bacteriophage could induce a strong immune response because of its ability to start a Toll-like receptor-mediated immune response, after its localization in LAMP-1 compartments, where the TLR9 receptor resides. Moreover, we propose that the bacteriophage, using an endosomal escape mechanism [90] (eukaryotic viruses, for instance, use this mechanism [91]), can leave the LAMP-1 compartments intact or partially dismantled, and can interact with cytoplasmic danger sensors, finding also into the cytosol the proteins able to start the DC maturation and downstream immune activation. Looking at the improved immune activation obtained when the bacteriophages are target to a specific receptor it is recommended to foresee targeting delivery of these nanoparticles in designing new generation vaccines.

## 5. General Considerations about Bacteriophages as Pharmaceutics

In experimental animal models, filamentous bacteriophage nanoparticles have been tested by various way of administration.

Engineered virions were administered subcutaneously, intramuscularly, intraperitoneally, intratumorally [75], intravenously [92] intranasally [93], and also intragastrically [94].

The injection of bacteriophages introduces them into the circulatory stream, regardless of the route of administration and distributes phages rapidly throughout the body [95].

Although most of the virions remain in circulation and are slowly cleared from blood with a half-life of 4.5 h, after 15 min the phages are already found in the liver, and after one hour they can be found practically in all the organs including spleen, lung, muscle, pancreas, and brain [96]. Phage particles remain trapped in the spleen for up to 7 days, while in the other organs are cleared more rapidly through the reticuloendothelial system, or extravasate to sub-endothelial tissue, and after 48 h only a few residual phages in these organs are found [97].

Indeed, a remarkable feature of filamentous phages is their ability to penetrate the biological barriers, including the blood–brain barrier [98]. Intranasally administered recombinant bacteriophages have been found both in the olfactory bulb and in the hippocampus region up to one year after administration without toxic effect, promoting research for the use of filamentous bacteriophages as therapeutics against Alzheimer’s Disease [33,99], Experimental Autoimmune Encephalomyelitis [100], Parkinson’s disease, and to select peptides binding vascular endothelium or tumor cells in a model of glioblastoma [101]. Convection-enhanced delivery of M13 bacteriophage to the central nervous system (CNS) in the primates demonstrates that bacteriophage penetration in the CNS was an active axonal transport process that should enhance distribution of virions in the brain [102].

Importantly, the modifications in the phage coat may alter the pharmacokinetics of bacteriophages: In fact, the succinylation or lactosylation of bacteriophages [97], as well as the display of targeting molecules, can enhance the uptake of the virions, reducing serum half-life [103].

Overall, although the administration of filamentous phages in vivo in humans requires in-depth and comprehensive investigations, there is evidence in the literature that bacteriophage injection in humans does not cause adverse effects.

In the 2014 clinical trial on multiple myeloma, based on phage idiotype vaccination, the cancer patients received six intradermal injections of bacteriophages at a dose level of 1  ×  10^11^–2.5  ×  10^12^ phages. The phage infusion was well tolerated by all the patients, who only developed mild and transient side effects, such as injection site skin irritation and flu-like symptoms [54]. All patients also developed serum anti phage antibodies, mainly IgG1 and IgG3 isotypes after 28 days from phage injection.

In a study to select tumor-binding antibodies in cancer patients with no pre-existing anti-phage antibodies, 1 × 10^11^ TUs/kg phages were injected intravenously without significant adverse events including allergy reactions during the infusion and up to 8-week follow up period [66].

In a previous in vivo panning study, the same authors show established toxicity profile of repeated doses of different types of phage-displayed libraries (random peptide and scFv libraries) administrations, demonstrating the safety of bacteriophages, with particular regard to the development of an anti-phage antibody response. All enrolled patients developed significant levels of serum IgG against filamentous bacteriophages in the 3–8 weeks following the phage administrations, but not during the one-week period when serial phage injections were performed [65]. Indeed, being a foreign virus, it is not surprising that the phage induces a strong anti-phage antibody response in the body after repeated administrations, also in the absence of adjuvant. Due to the peculiar structure of the phage, composed mainly of tightly packed pVIII proteins, the antibody response is restricted to 12 amino acids of the pVIII proteins that are surface-exposed, and to the more external N1-N2 domains of the pIII protein, which, being present in only five copies per virion, is not very immunogenic, allowing an antibody response more focused on exogenous polypeptides exposed on engineered virions [104].

Another potential advantage of using Ff phages as pharmaceutics is their high stability. pVIII recombinant bacteriophages show thermostability similar to wild type phages, being stable for greater than six months at room temperature. Phages can also resist at high temperatures, being still active after up to 5 weeks at 50 °C, and three days at 76 °C [105]. Exposure of filamentous phages to organic solvents [106], protein denaturants (i.e., guanidine chloride, urea, and dithiothreitol) [107] and proteases (trypsin, chymotrypsin, and proteinase K) did not cause loss of efficacy in infectivity and activity [108].

The stability of filamentous phages has also been examined in perspective for their feasible use as a drug in different storage conditions. Trehalose, mannitol, sucrose, and PEG_6000_ were used as excipients to stabilize and preserve phage activity after freeze-drying and long-term storage [109].

Compared to other bacteriophages, Ff phages are extremely resistant, even in harsh environments as the gastrointestinal tract and can survive to a wide range of pH above 3 [110], and are resistant to enzymatic degradation with hydrolytic enzymes [71].

Therefore, orally administered peptide-displaying bacteriophages may be used to induce epitope-specific immune responses. Intranasal and intragastric administration of filamentous phages displaying HCV and HBsAg mimotopes induces both systemic and mucosal IgA in mice [100] thanks to bacteriophages’ ability to cross the intestinal wall and migrate to gut-associated lymphoid tissue (GALT).

In addition, the development of genetically engineered phage preparations with improved pH resistance [111] or encapsulated in hydrogel microspheres [112] or pH-responsive polymer [113], can further improve the stability of phage-based vaccines for oral administrations. Furthermore, the encapsulation of bacteriophages in appropriate polymeric nanoparticles can improve the stability of bacteriophages in extreme pH environments and modulate their slow release, increasing their effectiveness as therapeutic [114].

As previously mentioned, a peculiar characteristic of phage nanoparticles is their enormous versatility and, because of their properties, there is also the probability to foresee further improvements. In this context, we recently explored the possibility of immobilizing lipid molecules on their surface. Lipid and glycolipid molecules have shown the ability to modulate the immune system and, therefore, can be promising adjuvants in vaccination [115,116,117]. The ability to deliver lipid molecules by filamentous bacteriophages can drive to the development of new drugs simultaneously carrying high-density antigens together with molecules delivering the phage to specific cell subsets and the lipid adjuvant to activate and sustain an antigen-specific immune response.

In conclusion, here we summarized evidence that these nature-made nanoparticles are well suited to be used as a therapeutic tool (Table 3).

Actually, the pathway to see the bacteriophage as a device ready off the shelf for human therapy or prophylaxis is still long and paved by many stumbling blocks. Most research has been performed so far in academic laboratories, with little emphasis on patenting; the collaboration between academic and industrial research will be instrumental in the clinical development of phage-based immunotherapeutics.

A major obstacle to the therapeutic use of bacteriophages is their characteristic of being “living” organisms, able to interact with the endogenous microbiome, with outcomes that are unpredictable using animal models. Strategies to decrease any risks associated with the use of phage nanoparticles are currently being studied. Bacteriophages can be inactivated without changing their immunogenicity; phage-based vaccines formulated to include heat-inactivated or UV-inactivated virions might represent safer alternatives to live phage vaccines [118,119]. Shorter (50 nm × 6 nm) pIII-recombinant Ff-derived particles (fd-nano) that do not carry any viral gene or antibiotic resistance and cannot replicate within bacterial cells or integrate into the bacterial genome have also been proposed [120].

Altogether, filamentous bacteriophages may represent an optimal platform for immunotherapeutic formulations. However, large clinical trials are needed in order to establish their safety in humans.

## Figures and Tables

**Figure 1 pharmaceutics-11-00437-f001:**
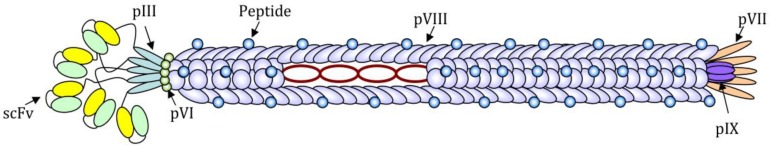
Schematic image of a filamentous bacteriophage nanoparticle engineered for the expression of a short antigenic peptide as a fusion with N-terminus of the pVIII protein and a single-chain antibody fragment (scFv) for the targeting, as a fusion with the N-terminus of the pIII protein. The circular single-strand DNA rich in CpG motifs can be recognized by PPR and acts as an adjuvant.

**Figure 2 pharmaceutics-11-00437-f002:**
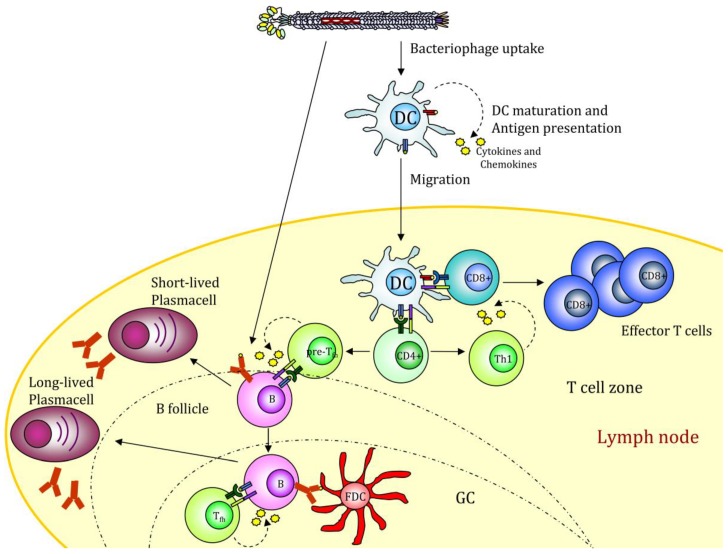
Filamentous bacteriophage nanovaccine can stimulate the humoral response or be taken up by antigen-presenting cells (APCs). The displayed antigenic peptides are processed and presented on MHCI and II molecules, leading to a CD4 and CD8 immune response. The presence of CpG sequences into the phage genome drives to APC maturation. GC (Germinal Center); FDC (Follicular Dendritic Cell). The figure depicts the possible scenario of target mediated fd internalization based on scientific results reported in [39,49].

**Figure 3 pharmaceutics-11-00437-f003:**
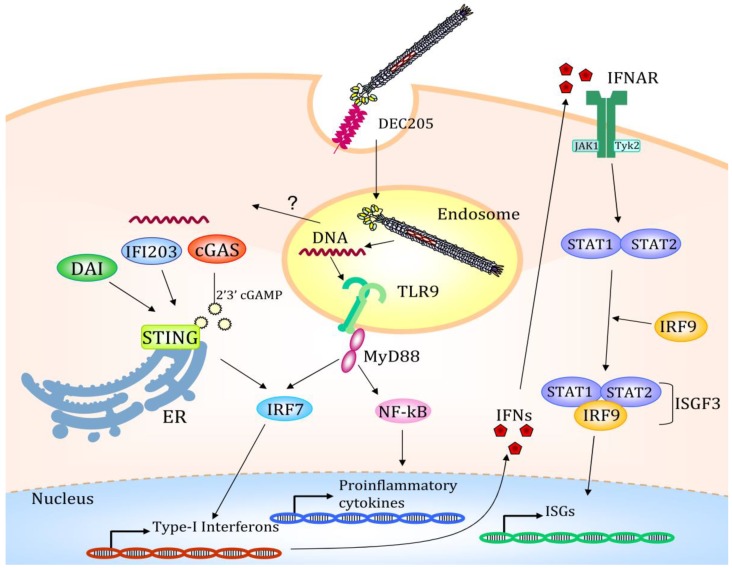
Once internalized via the DEC205 endocytic receptor, the filamentous bacteriophage is directed to the late endosomal compartments. Here the bacteriophage coat is degraded, and the CpG-rich DNA sequence starts the Toll-like receptor 9 (TLR9) mediated immune response with the activation of several transcription factors. A possible mechanism of endosomal escape is depicted and indicated with a question mark. Partially degraded viral DNA translocated to the cytosol starts a STING-mediated transcription activation. Type I IFNs promotes the STAT1/STAT2/IRF9 association to form the ISGF3 transcription factor regulating the transcription of IFN-stimulated genes (ISGs); IFNAR (Interferon-alpha/beta receptor). The figure depicts the possible scenario of target mediated fd internalization based on scientific results reported in [64,78].

**Table 1 pharmaceutics-11-00437-t001:** B and T epitopes expressed on filamentous bacteriophages and mentioned in the text.

Epitope	Protein	Sequence	Reference
Fba1	Fructose-bisphosphate aldolase 1 of *C. albicans*	YGKDVKDLFDYAQE	[24]
SE-CA-HSP90	Heat shock protein 90 of *C. albicans*	DEPAGE	[25]
HSP90_386–391_	Heat shock protein 90 of *C. albicans*	LKVIRK	[42]
Sap2_386–390_	Secreted aspartyl protease of *C. albicans*	VKYTS	[26,42]
Sap2_382–394_	Secreted aspartyl protease of *C. albicans*	SLAQVKYTSASSI	[43]
Gp70	Glycoprotein Gp70 of *S. globosa*	KPVGHALLTPLGLDR	[27]
Aβ 3–6	beta-amyloid peptide	EFRH	[31,32]
Aβ 2–6	beta-amyloid peptide	AEFRH	[33]
Pep23	Reverse transcriptase (RTase) of HIV-1	KDSWTVNDIQKLVGK	[39,40,44,45]
RT2	Reverse transcriptase (RTase) of HIV-1	ILKEPVHGV	[39,44,45]
p128, p30	Pp65 protein of Human Cytomegalovirus	EFFWDANDIYRIF, PLKMLNIPSINVHHY	[41]
S2_8–39_	Hepatitis B surface antigen (HBsAg)	IPQSLDSWWTSL	[46]
OVA_257–264_	Chicken egg ovalbumin	SIINFEKL	[47,48,62,64]
MAGE-A1_161–169_	Melanoma-associated antigen (MAGE)-A1	EADPTGHSY	[51]
MAGE-A3_271–279_	Melanoma-associated antigen (MAGE)-A3	FLWGPRALV	[53]
MAGE A10_254–262_	Melanoma-associated antigen (MAGE)-A10	GLYDGMEHL	[53]
P1A_35–43_	P1A of murine mastocytoma tumor P815	LPYLGVLVF	[52]
PA8	Amastigote surface protein 2 of T.cruzi	VNHRFTLV	[61]
TSKB20	Trans-sialidase of T.cruzi	ANYKFTLV	[61]

**Table 2 pharmaceutics-11-00437-t002:** Genes ^1^ overexpressed in dendritic cells (DC) treated with DEC205 targeted fd bacteriophage in comparison to untargeted bacteriophage.

***Transcription Factors***	
***Stat1***	Signal transducer and activator of transcription 1 STAT family members form homo- or heterodimers when phosphorylated by the receptor-associated kinases in response to cytokines. then they translocate to the cell nucleus acting as transcription activators.
***Stat2***	Signal transducer and activator of transcription 2 STAT family members forms homo- or heterodimers when phosphorylated by the receptor-associated kinases in response to cytokines. then they translocate to the cell nucleus acting as transcription activators.
***Irf7***	Interferon regulatory factor 7 Plays a role in the transcriptional activation of virus-inducible cellular genes, including interferon beta chain genes.
***Irf9***	Interferon regulatory factor 9 Transcription factor. IRF9 associates with the phosphorylated STAT1:STAT2 dimer to form a complex termed ISGF3 transcription factor. ISGF3 binds to the ISRE to activate the transcription of IFN-stimulated genes.
***Hif1alpha***	Hypoxia inducible factor 1, alpha subunit This gene encodes the alpha subunit which, along with the beta subunit, forms a heterodimeric transcription factor that regulates the cellular and developmental response to reduced oxygen tension.
***Nucleic Acid Binding Proteins***	
***Tmem***	Transmembrane protein 173 STING STING induces type I interferon production when cells are infected with intracellular pathogens, such as viruses, mycobacteria and intracellular parasites.
***Mb21d1***	Cyclic GMP-AMP Synthase Coding for cGAS protein: catalyzes the synthesis of cyclic guanosine monophosphate-adenosine monophosphate [cGAMP] after binding DNA in the cytoplasm.
***Nod1***	Nucleotide-binding oligomerization domain containing 1 Intracellular sensor system component.
***Ddx58***	DEAD [Asp-Glu-Ala-Asp] box polypeptide 58 RNA helicase It is involved in viral double-stranded [ds] RNA recognition and the regulation of immune response.
***Zbp1***	Z-DNA binding protein 1 Involved in the innate immune response. Binds foreign DNA and induces type-I interferon production.
***Eif2ak2***	Eukaryotic translation initiation factor 2-alpha kinase 2 dsRNA-dependent serine/threonine-protein kinase, induced by interferons. Involved in the innate immune response to viral infections.
***Antiviral Proteins***	
***Ifit1***	Interferon-induced protein with tetratricopeptide repeats 1 Involved in the cellular response to cytokine, bacteria, and virus
***Ifit3***	Interferon-induced protein with tetratricopeptide repeats 3 Inhibitor of cellular as well as viral processes
***Ifi203***	Interferon activated gene 203 Involved in the cellular response to interferon-beta.
***Ifi205***	Interferon activated gene 205 Involved the cellular response to interferon-beta and bacteria.
***Oas2***	2’-5’ oligoadenylate synthetase 2 Induced by interferons. Synthesize 2’,5’-oligoadenylates [2-5As] starting from adenosine triphosphate.
***Oas3***	2′-5′ Oligoadenylate Syntase Induced by interferons: catalyzes the 2’, 5’ oligomers of adenosine to bind and activate RNase L.
***Oas1g***	2’-5’ oligoadenylate synthetase 1G Involved in the defence response to viruses, the negative regulation of viral genome replication, and the regulation of ribonuclease activity.
***Trex1***	Three prime repair exonuclease 1 Metabolize DNA fragments of retroviral origin, including L1, LTR and SINE elements.
***TLR Pathway***	
***Tlr9***	Toll-like receptor 9 Binds DNA present in bacteria and viruses, and triggers signaling cascades, leading to a pro-inflammatory cytokine response.
***Unc93B***	Unc-93 homolog B1 TLR signaling regulator, involved in the innate immune response. Defects in the protein predispose to hypersensitivity to infections.
***MyD88***	Myeloid Differentiation primary response protein Adapter involved in the Toll-like receptor and IL-1 receptor signaling pathway in the innate immune response.
***Cytokine and Chemokine Ligands***	
***Il1b***	IL-1 b Proinflammatory cytokine.
***Cxcl10***	Chemokine [C-X-C motif] ligand 10 Involved in the defence response to viruses, the negative regulation of cell differentiation, and response to bacteria.
***Ccl7***	Chemokine [C-C motif] ligand 7 Involved in several processes, including the G protein-coupled receptor signaling pathway, the cellular response to cytokines, and leukocyte chemotaxis.
***Cxcl12***	Chemokine [C-X-C motif] ligand 12 Ligand for the G-protein coupled chemokine [C-X-C motif] receptor 4.
	Proteasome activator
***Psme1***	Proteasome activator subunit 1 (PA28 alpha) Proteasome activator that strongly increases the maximal velocity of the hydrolytic reaction and decreases the concentration of substrate required for cleavage
***Psme2***	Proteasome activator subunit 2 (PA28 beta) Proteasome activator that strongly increases the maximal velocity of the hydrolytic reaction and decreases the concentration of substrate required for cleavage
***ISGilation and Ubiquitination***	
***Isg15***	ISG15 ubiquitin-like modifier Ubiquitin-like protein conjugated to intracellular target proteins upon activation by interferon-alpha and interferon-beta
***Isg20***	Interferon stimulated gene 20 RNA binding activity and nuclease activity
***Transmembrane Receptor***	
***CD69***	Calcium dependent lectin superfamily type II transmembrane receptors

^1^ Selected genes with increased expression are reported. Function corresponds to the function of the protein encoded by the indicated gene as reported by MGI (Mouse Genome Informatics; http://www.informatics.jax.org/).

**Table 3 pharmaceutics-11-00437-t003:** Immunotherapeutic applications of filamentous bacteriophages.

Cargo	Display Format	Immune Response	Species	References
Cathepsin L mimotope fasciola hepatica	Recombinant pIII	Humoral response	*Capra hircus*	[12]
H5N1 influenza virus	Recombinant pIII	Humoral response	*Homo sapiens*	[16]
Fba1 *Candida albicans*	Recombinant pIII/Recombinant pVIII	Humoral/Cellular response	*Mus musculus*	[24]
HSP90 *Candida albicans*	Recombinant pVIII	Humoral/Cellular response	*Mus musculus*	[25,42]
Sap2 *Candida albicans*	Recombinant pVIII	Humoral/Cellular response	*Mus musculus*	[26,43]
Gp70 *Sporothrix globosa*	Recombinant pIII	Humoral/Cellular response	*Mus musculus*	[27]
beta-amyloid	Recombinant pVIII	Humoral response	*Mus musculus*	[30,31,32,33]
Reverse transcriptase HIV-1	Recombinant pVIII	Cellular response	*Homo sapiens/Mus musculus*	[39,44,45]
pp65 CMV	Recombinant pVIII	Cellular response	*Homo sapiens*	[41]
Ovalbumin	Recombinant pVIII	Cellular response	*Homo sapiens*	[48,61,62]
MAGE-A1/MAGE-A3/MAGE-A10 melanoma antigen	Recombinant pVIII	Cellular response	*Mus musculus*	[51,52,53]
Multiple myeloma idiotype	Chemical link	Humoral/Cellular response	*Homo sapiens*	[54]
P1A mastocytoma antigen	Recombinant pVIII	Cellular response	*Mus musculus*	[52]
Her2-Neu cancer antigen	Recombinant pVIII	Cellular response	*Mus musculus*	[55]
scFv anti-Carcinoembryonic antigen (CEA)	Recombinant pVIII	Cellular response	*Mus musculus*	[57]
PA8, TSKB20 Trypanosoma cruzi	Recombinant pVIII	Cellular response	*Mus musculus*	[61]
scFv anti- DEC205 receptor	Recombinant pIII	Innate immune response	*Mus musculus*	[62,64]
alpha-galactosylceramide	Chemical link	Cellular response	*Mus musculus*	[75]

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
