# Peer review of "Arming Filamentous Bacteriophage, a Nature-Made Nanoparticle, for New Vaccine and Immunotherapeutic Strategies"

_pharmaceutics, 2019, doi:10.3390/pharmaceutics11090437_

Round 1
Reviewer 1 Report
The review article titled “Arming filamentous bacteriophage, a nature-made nanoparticle, for new vaccine and immunotherapeutic strategies” addresses interesting topics in the pharmaceutical field.
I am in favor of publishing the manuscript after the authors make some adjustments that I consider important.
1) English must be reviewed by a native.
2) The authors should describe the methodology used in the search for the articles that were used in the paper. Search filters such as year and keywords should be clearly described.
3) In the topic involving the production of nanoparticles the authors should describe the types of particles used as well as specify the advantage of the system.
4) Item 6 patents can be removed, or authors should add explanatory text of it.
Author Response
Reviewer 1: The review article titled “Arming filamentous bacteriophage, a nature made nanoparticle, for new vaccine and immunotherapeutic strategies” addresses interesting topics in the pharmaceutical field. I am in favor of publishing the manuscript after the authors make some adjustments that I consider important.
1) English must be reviewed by a native.
English has been reviewed.
2) The authors should describe the methodology used in the search for the articles that were used in the paper. Search filters such as year and keywords should be clearly described.
We have now inserted the following footnote at the end of the text:
As literature search parameters, we at first performed a search to identify articles with keywords filamentous bacteriophage and phage-display and vaccines, filamentous bacteriophage and cytotoxic response, filamentous bacteriophage and cellular response, filamentous bacteriophage and antibodies, in the title, abstract, keywords, topic. Representative publications regarding phage-based vaccines and immunotherapeutic strategies were selected and then citing publications were searched, to identify recent developments. Quoting of reviews was limited and we left out articles on the same topic from the same author.
3) In the topic involving the production of nanoparticles the authors should describe the types of particles used as well as specify the advantage of the system.
We have modified the section 1 according to the reviewer suggestion (lane 86-94).
4) Item 6 patents can be removed, or authors should add explanatory text of it.
The item 6 has been removed.
Reviewer 2 Report
The authors have chosen an interesting topic and reviewed filamentous bacteriophages extensively in the current manuscript. This review may interest diverse research audience and adds good knowledge to the current literature. However, authors need to make some significant changes to the article before considering it for a publication.
1. Section 5 is incomplete, and the authors need to focus on this section. Pharmaceutical aspects like formulation design considerations, suitability of routes of administration, stability, pharmacokinetics, foreign body responses and adverse effects need to be reviewed extensively in this section.
2. Conclusions section is weak and does not discuss any future outlook for this delivery system. Authors need to give some insights on the state-of-the-art design methodologies required to improve these systems and also directions for future research.
3. Author should consider adding a table showcasing various filamentous bacteriophage delivery systems used for vaccine, immunotherapy and drug delivery purposes from the literature.
Author Response
Reviewer 2: The authors have chosen an interesting topic and reviewed filamentous bacteriophages extensively in the current manuscript. This review may interest diverse research audience and adds good knowledge to the current literature. However, authors need to make some significant
changes to the article before considering it for a publication.
We thank the reviewer for acknowledging the interest for the topic treated in this review.
Section 5 is incomplete, and the authors need to focus on this section. Pharmaceutical aspects like formulation design considerations, suitability of routes of administration, stability, pharmacokinetics, foreign body responses and adverse effects need to be reviewed extensively in this section.
We have written a new section 5 and more extensively addressed the points raised by the reviewer.
Conclusions section is weak and does not discuss any future outlook for this delivery system. Authors need to give some insights on the state-of-the-art design methodologies required to improve these systems and also directions for future research.
In agreement with the reviewer suggestion we also addressed these aspects in the revised section 5.
Author should consider adding a table showcasing various filamentous bacteriophage delivery systems used for vaccine, immunotherapy and drug delivery purposes from the literature.
As suggested by the reviewer we have inserted a Table (table 2) to list the filamentous bacteriophage delivery strategies reported in the literature.
Reviewer 3 Report
The review is concise and presents current knowledge about the utilization of filamentous bacteriophages in the field of drug delivery systems. I recommend to accept the manuscript in the present form - however, the title should be without the dot.
Author Response
The review is concise and presents current knowledge about the utilization of filamentous bacteriophages in the field of drug delivery systems. I recommend to accept the manuscript in the present form - however, the title should be without the dot.
We thank the reviewer for this comment. The dot has been removed.
Reviewer 4 Report
In this review article, Sartorius et al. have provided a survey on filamentous bacteriophage and made a relatively in-depth analysis of their pharmaceutical applications. The reviewed topic would be definitely interest to the interdisciplinary readership of pharmaceutics, including the researchers devoted to pharmacy, bioscience, and medicine. However, too many articles about the vaccine and immunotherapeutic of filamentous bacteriophage have been published, although they have not added these keywords in titles. Therefore, this manuscript covers less exciting and novel works, which may not provide a constructive guide for the researchers. Besides, the layout and structure of this manuscript are not well designed and organized. Consequently, I would recommend the acceptance of this review article in Pharmaceutics after the major revisions as follows. 1. As mentioned by the authors, the application of filamentous bacteriophage in medical field has been extensively reviewed. Thus, what are the innovations and emphases of this review? Long texts are not related to the filamentous bacteriophage in the section of Abstract and Introduction, which will affect the expression of theme and purpose. 2. Numerous essential and representative studies are not presented in this article. It is suggested that the author update the references and exhibit the latest research results in the form of photo compilation, so that readers might understand and grasp the scientific research trends more easily. 3. In Table 1, the authors simply list the overexpressed genes and their functions without summarizing and combing them. Besides, the relevant illustrations lack the corresponding references to support. 4. In the summary and outlook section, neither the possible problems nor the future development have been suggested, which reduces the reference value of this article. In review articles, authors must put forward their own viewpoints and suggestions after summarizing other latest research works. Unfortunately, it is seldom mentioned in this review. 5. In Line 373, the sentence of “Data obtained by RNAseq on DCs derived from the bone marrow of C57Bl/6 mice cultured with LPS-purified fd bacteriophages for 20 hours showed significant gene deregulation: in a stringent analysis of differentially expressed genes, we reported that genes involved in the bacterial invasion pathway, TNF signaling pathway, and focal adhesion pathway were upregulated” is directly given, what are cited literatures? Besides, the authors should provide more details of different filamentous bacteriophage on gene expression according to some references, then compare and discuss them. 6. There are many inconsistencies in the annotation, such as fdAD(2–6) (Line 196) and MAGE-A1(161-169) (Line 254), which make the article confused and cause reading obstacles.
Author Response
In this review article, Sartorius et al. have provided a survey on filamentous bacteriophage and made a relatively in-depth analysis of their pharmaceutical applications. The reviewed topic would be definitely interest to the interdisciplinary readership of pharmaceutics, including the researchers devoted to pharmacy, bioscience, and medicine. However, too many articles about the vaccine and immunotherapeutic of filamentous bacteriophage have been published, although they have not added these keywords in titles. Therefore, this manuscript covers less exciting and novel works, which may not provide a constructive guide for the researchers. Besides, the layout and structure of this manuscript are not well designed and organized. Consequently, I would recommend the acceptance of this review article in Pharmaceutics after the major revisions as follows.
As mentioned by the authors, the application of filamentous bacteriophage in medical field has been extensively reviewed. Thus, what are the innovations and emphases of this review? Long texts are not related to the filamentous bacteriophage in the section of Abstract and Introduction, which will affect the expression of theme and purpose.
It is known that current clinical and industrial interest on the use of bacteriophages in therapy mainly concerns the application of lytic phages. Filamentous bacteriophages may as well have a potential translational interest that has been object of reviews. However, the latest findings on this topic and a wide array of potential applications of filamentous bacteriophage have not been exhaustively covered so far in recent review papers. For this reason, we considered of interest for the readers to review the main and most recent aspects on the potential usage of filamentous bacteriophages: a field in which our group has been working for the last two decades. As suggested by the reviewer, we cut in the introduction section part of the text not related to the filamentous bacteriophage.
Numerous essential and representative studies are not presented in this article. It is suggested that the author update the references and exhibit the latest research results in the form of photo compilation, so that readers might understand and grasp the scientific research trends more easily.
We have now clarified in the footnote the criteria established by us for literature reviews. Applying these criteria, we made an effort to cover the main literature reports to the best of our knowledge. We are aware that some articles may have escaped our criteria and thus we are sorry if we missed to quote some articles related to the topic object of this review. We added table 2 to list the representative studies reported in the literature.
In Table 1, the authors simply list the overexpressed genes and their functions without summarizing and combing them.
We have reformatted the table organizing the reported genes by function.
Besides, the relevant illustrations lack the corresponding references to support.
We added the reference in figure legend 2 and 3
In the summary and outlook section, neither the possible problems nor the future development have been suggested, which reduces the reference value of this article. In review articles, authors must put forward their own viewpoints and suggestions after summarizing other latest research works. Unfortunately, it is seldom mentioned in this review.
We hope to have addressed the points raised by the reviewer in the revised section 5.
In Line 373, the sentence of “Data obtained by RNAseq on DCs derived from the bone marrow of C57Bl/6 mice cultured with LPS-purified fd bacteriophages for 20 hours showed significant gene deregulation: in a stringent analysis of differentially expressed genes, we reported that genes involved in the bacterial invasion pathway, TNF signaling pathway, and focal adhesion pathway were upregulated” is directly given, what are cited literatures?
We have now added the corresponding reference.
Besides, the authors should provide more details of different filamentous bacteriophage on gene expression according to some references, then compare and discuss them.
To the best of our knowledge, the only reports of differentially regulated gene expression analysis of BMDC challenged with fd bacteriophages are the ones discussed in the text. We have now modified the text in order to better describe the results reported (lane 487-494).
There are many inconsistencies in the annotation, such as fdAD(2–6) (lane 196) and MAGE-A1(161-169) (Line 254), which make the article confused and cause reading obstacles.
We have changed “fdAD(2–6)” in lane 232 with the AEFRH epitope and inserted the corresponding amino-acid sequence for MAGE-A1(161-169) : EADPTGHSY,(lane 318, 320) in order to smooth the reading obstacles.
Round 2
Reviewer 2 Report
Authors have addressed the comments adequately.
Author Response
We thank the reviewer for the useful comments
Reviewer 4 Report
The revision is ready for publication.
Author Response

(The authors gave the same response as above.)
